# Diffeomorphic Image Registration using Lipschitz Continuous Residual Networks

**Ankita Joshi**                                                    ANKITA.JOSHI25@UGA.EDU
*Department of Computer Science, University of Georgia, Athens, Georgia, USA*

**Yi Hong**[*]                                                        YI.HONG@SJTU.EDU.CN
*Department of Computer Science and Engineering, Shanghai Jiao Tong University, Shanghai, China*

## Abstract

Image registration is an essential task in medical image analysis. We propose two novel unsupervised diffeomorphic image registration networks, which use deep Residual Networks (ResNets) as numerical approximations of the underlying continuous diffeomorphic setting governed by ordinary differential equations (ODEs), viewed as a Eulerian discretization scheme. While considering the ODE-based parameterizations of diffeomorphisms, we consider both stationary and non-stationary (time varying) velocity fields as the driving velocities to solve the ODEs, which give rise to our two proposed architectures for diffeomorphic registration. We also employ Lipschitz-continuity on the Residual Networks in both architectures to define the admissible Hilbert space of velocity fields as a Reproducing Kernel Hilbert Spaces (RKHS) and regularize the smoothness of the velocity fields. We apply both registration networks to align and segment the OASIS brain MRI dataset. Experimental results demonstrate that our models are computational efficient and achieve comparable registration results with a smoother deformation field.

**Keywords:** Diffeomorphic image registration, residual networks, time dependent and stationary velocities.

## 1. Introduction

Image registration, a fundamental technique in many medical image analysis tasks, establishes global or local correspondences between a pair of images. Deformable registration methods desire to find a diffeomorphic deformation field, i.e. a smooth invertible deformation with a smooth inverse, to ensure the preservation of topology when deforming images. Traditional image registration algorithms like LDDMM (Beg et al., 2005) or SVF (Arsigny et al., 2006) successfully estimate diffeomorphic deformations to build dense correspondences between image pairs. While LDDMM achieves diffeomorphism by integrating a time dependent velocity field specified by a Riemennian metric, SVF achieves it by using a stationary velocity field in the corresponding Lie algebra. SVF thus works on the structure of the Lie group of diffeomorphic transformations, instead of the underlying Riemannian manifold (Kobatake and Masutani, 2017). However, these methods often involve computationally expensive high-dimensional optimization, causing practical challenges of providing time and memory efficient solutions for large-scale image studies.

---

[*] Corresponding author. This work was supported by NSF 1755970 and Shanghai Municipal Science and Technology Major Project 2021SHZDZX0102.

Recently, deep learning based approaches have been proposed to address the above challenges with highly parallelizable implementations executed on GPU, which make the deformation computations fast. Existing deep registration models tackle the efficiency challenge using supervised (Yang et al., 2017; Sokooti et al., 2017; Rohé et al., 2017) or unsupervised techniques (Dalca et al., 2018; Krebs et al., 2018). Supervised approaches maintain the diffeomorphic property, which is inherited from the classical diffeomorphic models like LDDMM, but they require the extra effort to obtain ground-truth deformations and their registration accuracy is limited by that of the obtained deformations. Inspired by the SVF framework, most of the unsupervised approaches have shown promising diffeomorphism and efficiency balanced registration results. However, because of the assumption of stationary velocity fields, these algorithms are limited in terms of flexibility and handling complex deformations, compared to dynamic systems considered in LDDMM. The SVF-based deep models are also limited by the computational costs of the scaling and squaring step (Arsigny et al., 2006; Higham, 2005; Mok and Chung, 2020) used for integrating stationary velocity-driven deformations, resulting in an integration at half-scale of the original dimensional size of inputs or downsampling original input size, or reducing model size of the used deep networks. The information loss during the approximation of underlying transformations or insufficient model capacity is a hindrance to the registration accuracy for high-resolution images, especially for 3D medical volumes.

In this paper, we propose two diffeomorphic image registration approaches inspired by the recent characterization of Residual Networks (ResNets) (Tai et al., 2017) as the numerical continuous flows of diffeomorphisms governed by ordinary differential equations (ODEs), based on both stationary and non-stationary velocity fields. This ResNet-based ODE solver makes it possible to perform the integration of the deformation mapping on the original scale of the input image with a time cost comparable to recent work like VoxelMorph (Dalca et al., 2018). Our contributions in the paper are summarized as follows:

- We propose two novel architectures using Lipschitz continuous ResNets, one with shared weights for numerical approximation of exponential diffeomorphic operators governed by an ODE with a stationary velocity field, and the other without shared weights for considering the flow of time-dependent (non-stationary) velocity fields.

- We conduct experiments for both models on the 3D OASIS brain MRI dataset. Results demonstrate that our framework with Lipschitz-continuous ResNets for both stationary and non-stationary velocity field integration provides comparable registration performance in terms of image matching while achieving smoother deformations.

## 2. Problem Formulation and Background

Given an image pair, a source image $I_0$ and a target image $I_1$, each of size $n_x \times n_y \times n_z$, the goal of diffeomorphic image registration is to determine a smooth deformation field $\phi : \Omega \to \Omega$ where $\Omega \subseteq R^{n_x \times n_y \times n_z}$ with a smooth $\phi^{-1}$, such that the image deformed from the source image, i.e., $\phi \cdot I_0$, is similar to the target image $I_1$. This problem is modeled as the minimization of an energy function below:

$$\mathcal{J}(\phi; , I_0, I_1) = \mathcal{D}(\phi(I_0), I_1) + \mathcal{R}(\phi), \tag{1}$$

where $\mathcal{D}$ is the matching term that ensures the deformed source image is similar to the target image, and $\mathcal{R}(\cdot)$ is a regularizer that controls the smoothness of the deformation $\phi$.

A diffeomorphic deformation field is determined by a smooth time-varying velocity field $v_t, t \in [0, 1]$, via the following ODE:

$$\frac{d}{dt}\phi_t = v_t \circ \phi_t, \quad \phi_0 = id, \tag{2}$$

with $\phi_t$ indicating the deformation at the time point $t$. $\phi_0$ is the identity map and $\phi_1$ is the deformation that transforms the source image $I_0$ to the target image $I_1$. Given the velocity fields $v_t$ and $\phi_0$, the computation of $\phi_1$ is the numerical integration of Eq. (2), given as $\phi_1 = \phi_0 + \int_0^1 v_t(\phi_t)dt$. On the other hand, as introduced in (Arsigny et al., 2006), the underlying Lie group of diffeomorphisms can be parameterized on stationary velocity fields, $(v_t = V, \forall t)$. This one-parameter subgroup is governed by the ODE:

$$\frac{d}{dt}\phi_t = V(\phi_t). \tag{3}$$

For the flow of a stationary velocity vector field in Eq. (3), the solution of $\phi(t)$ is represented as the exponential of the velocity $V$ given as:

$$\phi(t) = \exp(tV). \tag{4}$$

## 2.1. Related Work

Following the scaling and squaring methodology used to solve Eq. (4) (Arsigny et al., 2006; Higham, 2005) , most of the current unsupervised registration algorithms (Mok and Chung, 2020; Dalca et al., 2018; Krebs et al., 2018) use the integration of stationary velocity fields to produce diffeomorphic deformations. However, although each of these algorithms provide diffeomorphic transformations, they do not use time-dependent velocity fields and are less flexible for handling large deformations. Residual Networks (ResNets) have been proposed to be used for the task of image registration in place of the popular convolutional neural networks (Tai et al., 2017). Recently, some researchers use ResNets to learn deep multi-scale residual representations to boost the registration accuracy (Yang et al., 2021), others use the characterization of ResNets as numerical schemes of differential equations by relating the mapping blocks in the ResNets with the integration steps in the popular LDDMM algorithm (Ben Amor et al., 2021). The authors use the incremental mappings of ResNets to parameterize time-dependent affine velocity fields and follow the LDDMM framework to use a regularizer for summation over all kinetic energies of the system at all time-steps and apply their framework in the task of registration problems on shapes.

There are multiple works to provide insights into ResNets from an aspect of ODE/PDE (partial differential equation) and relate the incremental mapping (residual blocks) defined by ResNets as numerical schemes of differential equations used in diffeomorphic registration models, especially to deep LDDMM. Currently, multiple methods have been proposed based on using continuous optimization dynamics via neural ODEs (Chen et al., 2018) for image registration. In (Wu et al., 2021) authors use a novel multi-scale approach based on neural ODEs and a modality independent similarity measure for the task of image registration. However, they do not guarantee diffeomorphic deformations in their registration results. A

similar method to use neural ODEs is proposed in (Xu et al., 2021) using an optimization based strategy for registration, which is more relevant to the classical registration methods; however, they use stationary velocity fields for performing 3D image registrations. Recurrent neural networks have also been used in (Sandkühler et al., 2019) as a novel sequence-based framework to generate sequence-based transformations instead of directly estimating the final transformation in one step. They use an interpolating transformation model based on a fixed basis function. However, even though this model achieves global smoothness, it would hinder the preservation of local details that a dense model allows. Since the final transformation is based on the previous series of transformations, it highly depends on a good estimation of the initial transformation.

Alternatively, inspired by (Rousseau et al., 2020) that provides theoretical and computational insights of characterization of ResNet architectures as the numerical implementation of diffeomorphic continuous flows governed by ODEs, we use ResNets to solve Eq. (2) and (3), providing a framework for both stationary and time-dependent velocity field integration.

## 3. Registration-based Interpretation of ResNets

The mapping block in a Residual Network, which incrementally maps the embedding space onto a new unknown space, has the following form:

$$x_{l+1} = x_l + F(x_l, \theta_l), \tag{5}$$

where $x_l$ is the input to the $l^{th}$ residual unit and $\theta_l$ are the trainable weights associated with the $l^{th}$ residual unit. This incremental mapping points a striking similarity to diffeomorphic registration models, which tackles the registration problem by composing a series of incremental diffeomorphic mappings. Each mapping is close to the identity and integrates using the forward Euler method with a given initial value. Training a deep residual network is often viewed as a discretization of a dynamical system governed by the first-order ODE, where the network layers are viewed as time-steps and the network parameters, and $\theta_l$ are viewed as the control to optimize (Liu and Theodorou, 2019). The discrete paramterization of velocity field $v_t$ with the neural network based representations of ODEs in (2) and (3) can be considered as a linear combination of basis functions using ResNet mapping in Eq. (5):

$$v_{t+1} = v_t + F(v_t, \theta_t), \tag{6}$$

where each mapping block $t$ is viewed as a time-step, and $\theta_t$ represents the network parameters. Similarly, each residual unit of a ResNet is expected to implement the composition of a series of diffeomorphic mappings. Such a connection makes the function $F$ to be a parameterization of a deformation flow field. Moreover, residual units with shared weights are viewed as the numerical implementation of the exponential of velocity fields, which is the diffeomorphic operator governed by stationary velocity fields (Rousseau et al., 2020).

To this end, we use Residual Networks to formulate our image registration problem, with additional Lipschitz constraints (see Section 4) on the velocity fields for generating diffeomorphic deformations under an unsupervised scheme.

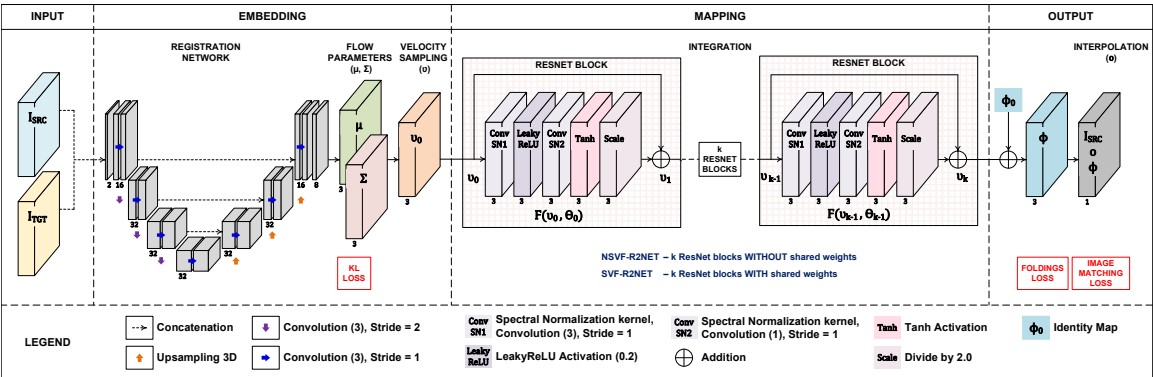

Figure 1: Overview of our proposed network.The ResNet block without shared weights corresponds to the time-varying velocity field, while the one with shared weights correspond to the stationary velocity field. Both models use k=7 ResNet blocks.

## 4. Proposed Methods

Figure 1 shows two proposed architectures for diffeomorphic image registration. We name the registration architecture based on Non-Stationary (time-varying) Velocity Fields as NSVF-R2Net (Registration Residual Network) and the one based on Stationary Velocity Fields as SVF-R2Net. Below are network designs shared by both architectures:

**Initial estimation of velocity field.** Firstly, for both architectures, we approximate the posterior probability parameters representing the velocity field mean and variance using an encoder-and-decoder scheme as used in (Dalca et al., 2018). We use the UNet architecture (Ronneberger et al., 2015), which takes in image pairs and outputs the mean $\mu$ and the variance $\Sigma$ for sampling a corresponding initial velocity field. Although such a scheme can be used for a wide range of representations, we let the estimated $v$ be a non-stationary velocity field that drives a diffeomorphic mapping $\phi$ through the ODE in Eq. (2) for NSVF-R2Net, and use it as a stationary one that specifies diffeomorphisms through the ODE in Eq. (3) for SVF-R2Net. This constitutes the parameter-estimating phase of both architectures.

**Integration of velocity fields.** As discussed before, we utilize ResNets as numerical schemes of differential equations and relate the incremental mapping defined by ResNets to diffeomorphic registration models. In this regard, we have a series of ResNet blocks without sharing weights (see Fig. 1), which compute time-dependent velocity fields. This constitutes as the integration component for NSVF-R2Net. Similarly, we utilize the same ResNet block but with shared weights to compute stationary velocity fields. This constitutes as the integration component for SVF-R2Net.

**Spectral normalization for Lispchitz continuity.** A number of works advocate the importance of Lipschitz continuity in assuring the boundedness of statistics (Yoshida and Miyato, 2017; Gouk et al., 2021). LDDMM defines an admissible Hilbert space of velocity fields with adequate smoothness conditions as an RKHS (Reproducing Kernel Hilbert Space). Under adequate assumptions on the velocity fields $v$, with a time-dependent Lipschitz constant integrable in time, $v \rightarrow \phi_1$ is a well-defined mapping into the space of time-dependent diffeomorphisms by the Cauchy-Lipschtiz theorem. A smoother $v$ yields

| Method | RMSE ($e^{-3}$) | #Foldings | Avg. Dice | Inference (per img. pair) | Memory (GB on GPU) |
|---|---|---|---|---|---|
| SyN | $\mathbf{1.08 \pm 0.00}$ | $47.70 \pm 145.14$ | 0.67 | 10 min (CPU) | - |
| VoxelMorph | $1.10 \pm 0.00$ | $51.43 \pm 83.76$ | **0.72** | 450ms+1440ms | 8.6 |
| SVF-R2Net | $1.54 \pm 0.00$ | $\mathbf{28.41 \pm 17.56}$ | 0.69 | **900ms** | **7.9** |
| NSVF-R2Net | $1.34 \pm 0.00$ | $\mathbf{27.23 \pm 47.44}$ | 0.70 | **900ms** | **8.5** |

Table 1: Summary of quantitative results for all algorithms: Mean squared error over all test images, mean number of locations with a non-positive determinant of Jacobian, mean Dice scores over all anatomical structures and subjects, the inference time for a pair of images and the memory required for training models on a single GPU.

a smoother $\phi_1$, as seen in (Younes, 2010) (Theorem 8.7). A recent work connects neural networks and RKHS (Bietti and Mairal, 2019) (Proof for Proposition 14), which shows convolutional neural networks with homogenous activation functions (tanh) fall under RKHS. Following this, we employ the method proposed in (Miyato et al., 2018) to enforce Lipschitz continuity of Residual blocks in both architectures, i.e., using the spectral normalization for each convolutional layer inside the residual units. This operation normalizes the spectral norm of the weight matrix $\mathcal{W}$ so that after the spectral normalization it satisfies the Lipschitz constraint $\delta(\hat{\mathcal{W}}_{SN}) = 1$, that is, $\hat{\mathcal{W}}_{SN} := \frac{\mathcal{W}}{\delta(\mathcal{W})}$. Having this condition implies that the Hilbert space of admissible velocity fields is an RKHS.

**Loss functions.** We use similarity metric $\mathcal{L}_{sim}$ to ensure good image matching. The Kullback-Leibler loss $\mathcal{L}_{KL}$ is used between the expected multivariate normal distribution $\mathcal{N}(0, 1)$ and the real latent distribution $\mathcal{N}(\mu, \gamma)$, which are the mean and variance of the learned initial velocity field. Lastly, $\mathcal{L}_{Jdet}$ is used to enforce the smoothness of the deformation by penalizing the total number of locations, where the Jacobian determinants $|\mathbb{J}(\phi(x))|$ are negative (Kuang and Schmah, 2019). The overall objective function is formulated as

$$\mathcal{L} = \lambda_1 \mathcal{L}_{sim}(\frac{1}{2\sigma^2 K}\|I_1 - \phi \circ I_0\|_2^2) + \lambda_2 \mathcal{L}_{KL}(\mathcal{N}(0,1), \mathcal{N}(\mu, \gamma)) + \lambda_3 \mathcal{L}_{Jdet}(0.5(|\mathbb{J}(\phi(x))| - \mathbb{J}(\phi(x)))),$$
(7)

where $K$ is the number of samples, and $\lambda_1$, $\lambda_2$ and $\lambda_3$ are the balancing weights, for which values of $[2, 1, 1]$ worked the best in the experiments, after trials over a few training cycles.

## 5. Experiments

We evaluate our framework for the task of brain MRI registration. All experiments for both architectures NSVF-R2Net and SVF-R2Net are done in 3D. Experiments for VoxelMorph, NSVF-R2Net and SVF-R2Net were performed using a Titan X GPU.

**Dataset.** We use the OASIS brain MRI dataset (Marcus et al., 2007; Hoopes et al., 2021), which contains T1-w MRI scans for 414 subjects preprocessed with skullstripping, bias-correction, registered and resampled into the freesurfer's talairach space. After preprocessing, each volume has dimensions of $160 \times 192 \times 224$. We divide these 414 subjects into sets of 264, 100 and 50 as our training, test, and validation groups. We then randomly pair images in each set and choose 350 pairs for training, 50 for validation, and 100 for testing.

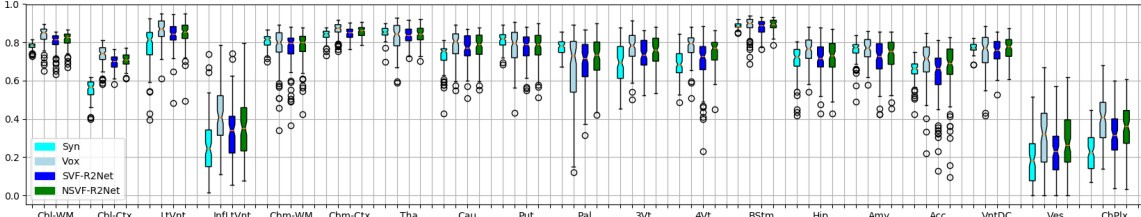

Figure 2: Dice scores for segmenting anatomical structures using SyN, VM, and ours.

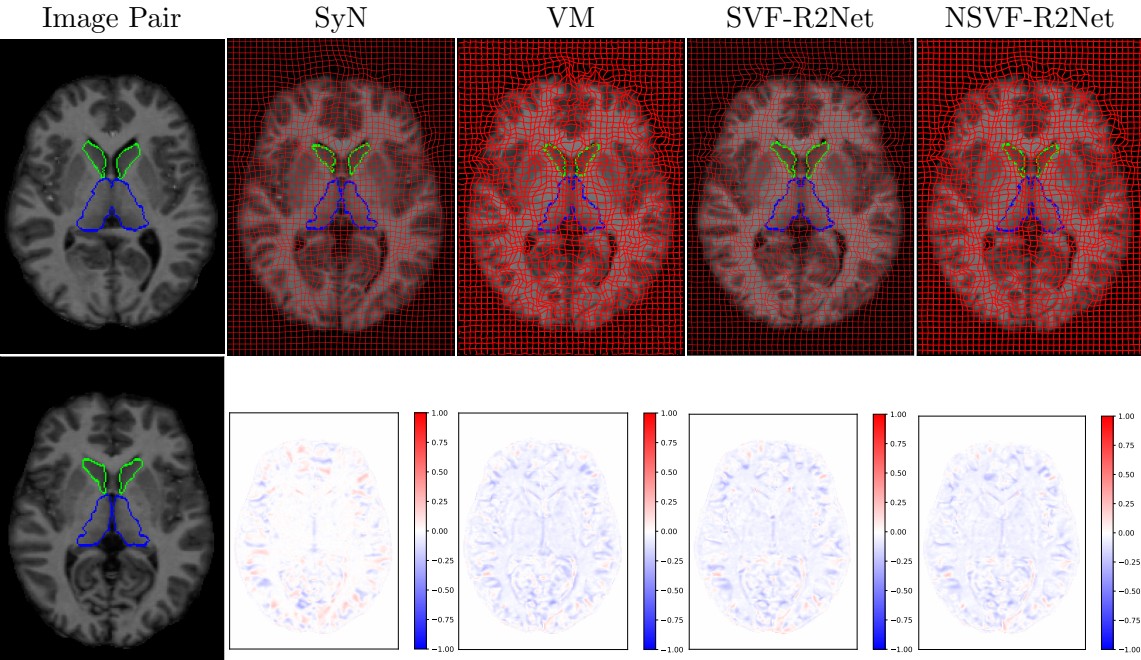

Figure 3: Registration comparison among SyN, VM (VoxelMorph), and our models. The deformation (red) and deformed mask (Caudate in green, Thalamus in blue) are overlaid on the deformed image, and its difference w.r.t the target is shown below.

**Baselines and Evaluation metrics.** We compare our models to the classical registration algorithm ANTs SyN (Avants et al., 2011) from the ANTsPy package with manually tuned parameters on a few training images and the deep learning based diffeomorphic version of VoxelMorph (VM) (Dalca et al., 2018) with their given default parameters. We evaluate the registration performance in terms of image matching by computing the intensity Root Mean Square Error (RMSE). We also evaluate the Dice score between the segmentation mask of the target image and the deformed mask of the source image, using the deformation field estimated for the input image pairs. We compute the average Dice score across 35 anatomical structures. To quantify the deformation smoothness, we count the number of its voxels with negative Jacobian determinants. In order to measure the resource utilization, we state the time and memory requirements for all algorithms.

## 6. Results and Discussion

The quantitative results for all algorithms are shown in Table 1. It can be seen that our methods (SVF-R2Net and NSVF-R2Net) achieve comparable results to those of VoxelMorph and SyN in terms of image matching and averaged segmentation Dice. Further improvement of image matching term is possible by increasing the weight of the image-matching loss during training, at the expense of increasing the number of foldings. NSVF-R2Net achieves slightly better performance than SVF-R2Net, which is probably due to the fact of increased flexibility due to non-sharing weights in the mapping block. This helps achieve better matching results due to better convergence during training. The Dice scores in Figure 2 show both architectures produced comparable Dice scores as VoxelMorph, across the anatomical structures, while doing better than SyN, in all regions. More importantly, compared to other algorithms our method produced significantly smoother and more regular deformations as shown by the number of foldings in Table 1 and the qualitative results in Fig. 3.

Regarding the computational cost, the amount of GPU memory utilized by both SVF-R2Net and NSVF-R2Net is less than that of VoxelMorph (see Table 1). Worth to mention that, VoxelMorph works on the integration of half-scale velocity fields, while our models work on full-scale resolutions. Futhermore, VoxelMorph takes about $450ms$ to estimate the velocity fields for an image pair; however, the output flow parameters from VoxelMorph have to be integrated and upsampled to match the original resolution of the input image. To obtain final scale deformation fields, the task of integrating (via scaling and squaring) and upsampling for a single output takes about additional 1.44 seconds. Our SVF-R2Net and NSVF-R2Net provide the final deformation field at the original image resolution in just one pass through the trained model, which takes less than *one* second. SyN can be performed only on CPU due to the unavailability of a GPU based implementation, which needs about 10 minutes. That is, our models are efficient in terms of both memory and inference time cost, which makes it possible to handle higher resolution image scans.

## 7. Conclusion

In this paper we have proposed two unsupervised diffeomorphic image registration models, i.e., SVF-R2Net and NSVF-R2Net, using time-dependent and stationary velocity fields, respectively, to drive deformations through ODEs. We utilize Lipschitz-continuous ResNets as numerical schemes of differential equations for generating diffeomorphic mappings. Our models show comparable quantitative and qualitative results in terms of image-matching with improved performance on the smoothness of the generated diffeomorphic deformation fields. Our architectures also show better resource utilization, which is memory and inference efficient in comparison to two popular diffeomorphic registration algorithms.

A possible future approach involves considering a multi-scale approach to improve the registration performance. Various other techniques of velocity estimation and operations other than spectral normalization to achieve smooth velocity fields could be considered to improve the overall registration performance. Another enhancement to the current approach could be the use of Invertible Residual Networks (Behrmann et al., 2019), which obtains a better estimate for the spectral norm than the current upper bound used in this work.

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

## Appendix A. Computational Cost

In this section, we study the computational cost of VoxelMorph (VM), SVF-R2Net and NSVF-R2Net. All experiments are carried out on a single Titan X GPU. SyN was not included in this experiment due to the unavailability of a GPU-based implementation.

Table 2 shows the training time and memory cost for varying sizes of input image resolutions starting from $64 \times 64 \times 64$ up to $224 \times 224 \times 224$. The public implementation of VoxelMorph was modified, in that, the integration of the velocity fields was done on the original input sizes instead of their default half size, in order for it to be a fair comparison with SVF-R2Net and NSVF-R2Net, which both work on original scale velocity fields. Our methods consistently need less memory compared to VM. More importantly, VM cannot handle the size of $192 \times 192 \times 192$ and $224 \times 224 \times 224$, but ours can. In each training iteration, SVF-R2Net and NSVF-R2Net take a little more time compared to VM.

In terms of how each algorithm handles scaling up of the input dimension, we ran experiments for the input size of $128 \times 128 \times 128$. While SVF-R2Net and NSVF-R2Net can handle up to 4 input image pairs on a single GPU, both taking 2 seconds each to train all 4 pairs in a single iteration, VoxelMorph can handle only 2 such image pairs and takes 2 seconds to finish a single iteration. This shows that the proposed architectures SVF-R2Net and NSVF-R2Net, can handle higher resolutions of images and scale-up better, compared with the baseline algorithm VoxelMorph.

| Dataset Size | VM | | SVF-R2Net | | NSVF-R2Net | |
| :---: | :---: | :---: | :---: | :---: | :---: | :---: |
| $n^3$ | Time(s) | Memory (GB) | Time(s) | Memory (GB) | Time(s) | Memory (GB) |
| 64 | 0.113 | 1.4 | 0.170 | 0.9 | 0.174 | 0.9 |
| 96 | 0.298 | 2.4 | 0.432 | 1.4 | 0.433 | 1.4 |
| 112 | 0.419 | 4.4 | 0.651 | 2.4 | 0.660 | 2.4 |
| 128 | 0.600 | 8.5 | 0.915 | 4.5 | 0.908 | 4.5 |
| 144 | 0.907 | 8.5 | 1 | 4.5 | 1 | 4.5 |
| 192 | – | – | 3 | 8.5 | 3 | 8.5 |
| 224 | – | – | 4 | 11.8 | 4 | 11.8 |

Table 2: Training Time (sec/iteration) and GPU Memory (GB) consumption for VM (VoxelMorph), and our models. All experiments are carried out on one input image pair, and the reported values are averaged over 10 runs.

## Appendix B. Spectral Normalization

In this section we provide background for the spectral normalization algorithm used in both proposed architectures and its implementation. In the work proposed by (Miyato et al., 2018), Algorithm 1, the authors use fast approximation by the power iteration method in order to replace the weights $W$ with $\delta(W)$, the largest singular value of $W$. This is done to avoid the added computational complexity of computing the singular value decomposition. For our work, we used the open source Keras implementation[1].

---

1. https://github.com/IShengFang/SpectralNormalizationKeras

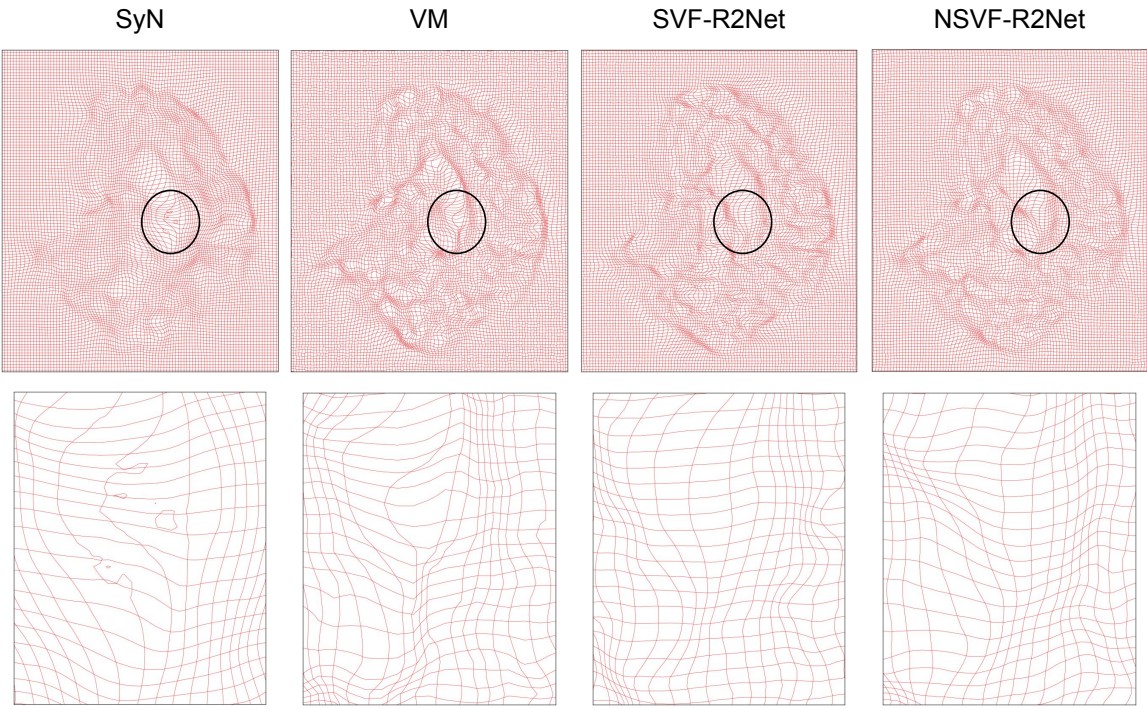

Figure 4: One sample showing deformations zoomed in the same region across SyN, VM, SVF-R2Net, and NSVF-R2Net.

## Appendix C. Smoothness Visualization

In this section, we show a few zoomed-in examples of the same regions within the generated deformations using all algorithms: SyN, VM, SVF-R2Net and NSVF-R2Net. It can be seen in Figure 4 and 5 that while SyN and VoxelMorph produce non-smooth deformations such as foldings or overly-compressed curves, SVF-R2Net and NSVF-R2Net always produce smooth deformations. It can also be seen that the background region is unsmooth for VoxelMorph, which is not the case with SyN, SVF-R2Net, and NSVF-R2Net.

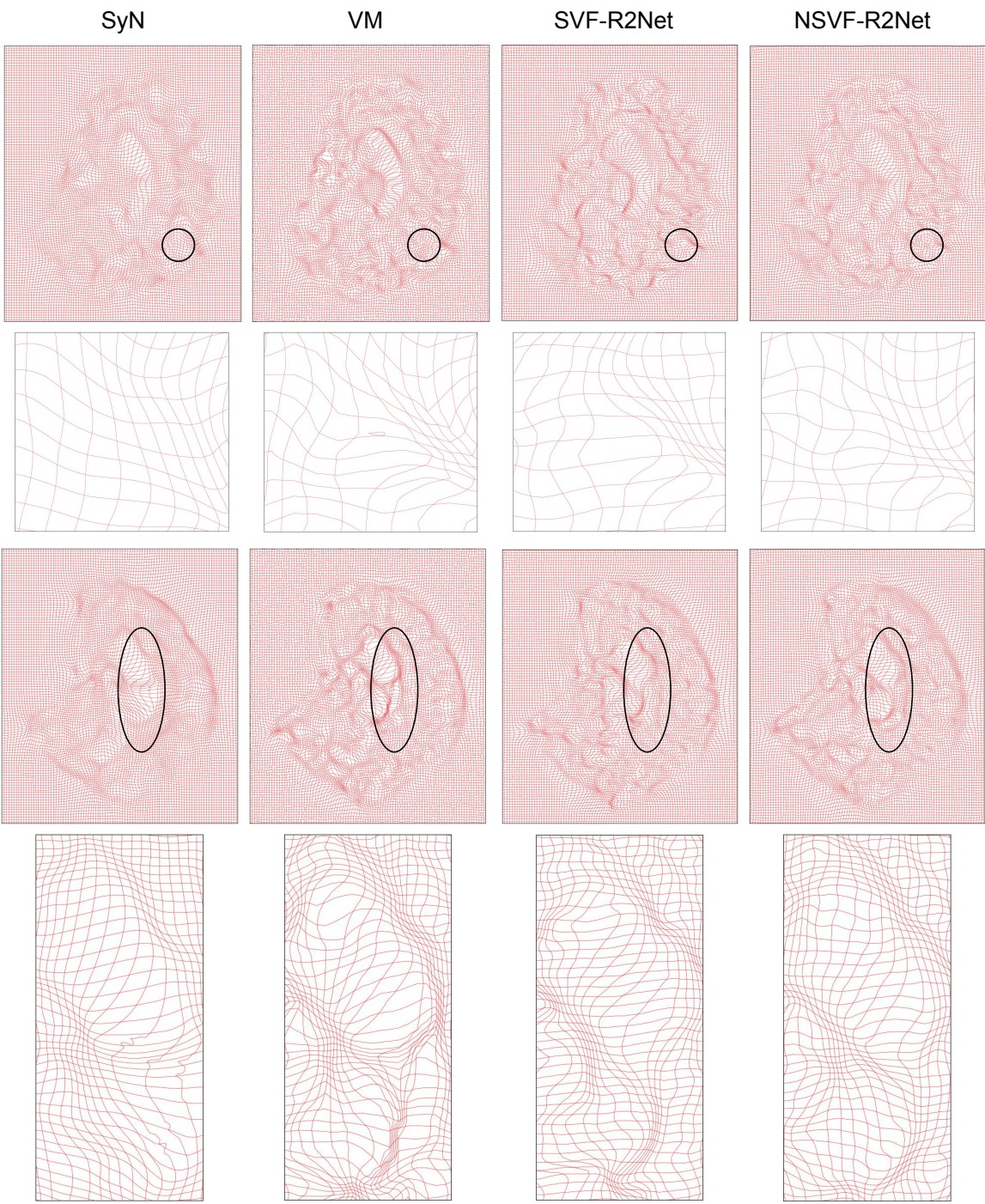

Figure 5: Another two samples showing deformations zoomed in the same regions across SyN, VM, SVF-R2Net, and NSVF-R2Net.

