# OpenReview forum: "Diffeomorphic Image Registration using Lipschitz Continuous Residual Networks."
_MIDL.io/2022/Conference — MIDL 2022_

### Official Review · Reviewer_aDjC · 2022-01-20

**Confidence:** 4
**Preliminary Rating:** 5
**Recommendation:** Oral, Poster

**Summary:**

The authors propose two new neural network based diffeomorphic image registration variants. Both variants work from the principle that residual networks can be considered, under certain circumstances as solving an ODE. As diffeomorphic deformation fields can be determined by an ODE, they authors propose solving this ODE with a residual network. By utilizing shared weights across residual blocks, they can achieve a stationary velocity field, otherwise the solution results in a time dependent velocity field.

**Strengths:**

The paper is well written with a thorough background and related work section. Providing computation requirements and run times is important to judge if the proposed model is viable in the wild, which the authors did.

**Weaknesses:**

Evaluation could have been done on a wider range of anatomies or more examples could have been provided. Given the models preform comparably to the baselines, more analysis into failure modes with examples to better distinguish the pros and cons of each algorithm and where they differ would have been interesting.

**Deanonymize Review:**

no

**Detailed Comments:**

Dataset: Though each dataset has distinct pairs, it would imply that a particular image may exist across the sets. Given the number of available scans, would it not be beneficial to split the dataset into patients then generate a larger number of pairs for training, validation, and testing?
Was the voxel morph baseline retrained or fine tunes on the OASIS data or was a pretrain model used?
Figure 3 is difficult to interpret, consider overlaying the images and enlarging them.
please fix “short title” issue


**Final Rating After The Rebuttal:**

5: Strong Accept

**Justification Of The Final Rating:**

I would like to thank the authors for updating the manuscript and addressing the few concerns I had. I am fully satisfied with the clarifications the authors have provided and my final decision remains the same.

**Paper Type:**

methodological development

**Questions To Address In The Rebuttal:**

Overall, the paper is well written and the approach in interesting, clarifications on the points made in the `detailed comments` section would be appreciated. More details on practically how to achieve Lipchitz continuous residual blocks would also make the paper more self-contained

**Special Issue:**

yes

---

### Official Review · Reviewer_rwHf · 2022-01-24

**Confidence:** 4
**Preliminary Rating:** 3
**Recommendation:** Poster

**Summary:**

This paper improves on previous methods on diffeomorphic image registration with deep learning by using Lipschitz residual networks. With benchmark datasets, they show similar accuracy to a few previous methods based on similar architectures and improved smoothness of the solution with faster numerical evaluations.

**Strengths:**

The paper is generally well written, with a few minor details which could be included to improve the exposition (see below). The exposition is clear and the method is well introduced and put into context.

**Weaknesses:**

I am not completely familiar with these ResNet architectures for registration, but the novelty of this paper appears somewhat small to me. If I understand it well, the inclusion of the Lipschitz normalisation is the only main novelty (apart from potential small details on the learning architecture), but, as viewed from the number of citations from the cited 2018ICLR paper on that method, it seems to have already become a classical and must use method. The used dataset is also one of the benchmark, but one would have like to see this method applied to some other dataset, where the smoothness of the matching matters more, or the resolution makes other methods too computationally intensive, etc...

**Deanonymize Review:**

no

**Detailed Comments:**

- The spectral norm sigma in eq 18 is not defined (and could be confused with the std sigma a few lines below).
- the quantity number of folding crucial to show that this work provides smoother matching is not defined (nor a paper with its definition cited)

**Final Rating After The Rebuttal:**

4: Weak Accept

**Justification Of The Final Rating:**

All good for me! The replies are satisfactory............................................................................... sorry for the dots, I didn't know what to write to fill in the 200 chars...

**Paper Type:**

both

**Questions To Address In The Rebuttal:**

- As one of the stated strengths is the increased smoothness of the result, it would be good to have a more thorough study on that. For example as a function of loss hyperparameters, for different datasets, using different measures of smoothness to better quantify it. Showing zooms of non-smooth regions with the previous method vs the same region with the current method, etc...

- As one of the stated strengths is the reduced computational cost, it would be good to have a more thorough study on that. For example, some plots of computational complexity with various parameters of the dataset, some more details on which part of the algorithm is faster, etc... With one data point on computational speed, it may be faster, but the scaling may be worse than other methods, so overall, not better, as stated. One would at least want to see the computational cost of subsampled or oversampled datasets to have a better assessment of the improvement in that regard.

**Special Issue:**

no

---

### Official Review · Reviewer_kLGt · 2022-01-24

**Confidence:** 3
**Preliminary Rating:** 4
**Recommendation:** Oral, Poster

**Summary:**

The paper presents two unsupervised diffeomorphic image registration models based on numerical integration of stationary and non-stationary velocity fields, respectively. The authors propose to use incremental mappings of Lipschitz continuous ResBlocks to solve the numerical integration of the velocity field to solve the ordinary differential equation that determines a diffeomorphic deformation field. This is motivated by the similarity of residual networks and the forward Euler method for numerically solving ODEs. Lipschitz continuity is enforced by spectral normalization of the weight matrices of the integrating ResBlocks.

The initial velocity field is sampled from the posterior distribution of a probabilistic UNet, similar to Dalca et al. (2018). The overall model is trained end-to-end in an unsupervised manner. The loss function comprises a voxel-wise similarity metric between the warped source and fixed target image, the negative log-evidence lower bound (KL between the variational posterior and the multivariate Normal prior), and penalization of negative Jacobian determinants. The presented method is experimentally evaluated on 3D MRIs of the brain and peforms similar to existing methods by means of registration accuracy and outperforms other methods by means of a smooth and diffeomorphic displacement field.

**Strengths:**

The paper presents an interesting approach to registration by exploiting the similarity between residual learning and nurmerical integration of velocity fields used to solve the ODE that governs diffeomorphic deformations. To further ensure smooth velocity fields, the authors link Lipschitz continuity of ResBlocks to RKHS and propose to use spectral normalization of the weight matrices. I think that the latter is the core contribution of this paper, enabling smoother deformation fields with less foldings (i.e., $ \text{det} J < 0 $). This is a clever approach to the otherwise used integration with scaling and squaring.

The paper is well written and clear, albeit densely packed with information.

**Weaknesses:**

The paper uses a probabilistic UNet as presented by Dalca et al. (2018) without any rationale. Dalca et al. (2018) assessed the accompanying velocity field uncertainty to make the method more robust and fail-safe and showed that uncertainty is low near structure boundaries. Using a variational Bayes approach to the velocity field here seems unnecessarily complicated and the added benefit is not clear to me.

After reading (Miyato et al., 2018), the implementation of the spectral norm seems non-trivial. However, no details on how the spectral norm is computed are given.

Moreover, the paper is densely packed with information and contains many assumptions to the reader that makes it--at least for me--somewhat hard to read. I had to review a lot of background information to fully grasp the presented method. Maybe this can be attributed to the 8 page limit and the paper would benefit from an extended version.

Finally, it seems that the code is not and will not be public. Public code is highly anticipated.

**Deanonymize Review:**

no

**Detailed Comments:**

Some minor comments:

* Last sentence of § 7 is unclear to me ("... which obtains a better estimate for the spectral norm than the current upper bound used in this work.").

**Final Rating After The Rebuttal:**

4: Weak Accept

**Justification Of The Final Rating:**

The core contribution is the use of Lipschitz continuous ResNets to integrate the velocity field in the context of an otherwise well-known registration framework. The link between ResNets and solving neural ODEs, as well as the use of the spectral norm for Lipschitz continuity are also well-known. However, I appreciate the combination of the existing building blocks to address diffeomorphic image registration and would like to see this paper discussed at MIDL 2022.

In the rebuttal, the authors answered most of my questions sufficiently. The rationale behind the variational Bayesian approach is still not 100 % clear to me. I think that the paper would benefit from an extended version. Overall, my rating remains.

**Paper Type:**

both

**Questions To Address In The Rebuttal:**

* What is the rationale behind the variational Bayes approach?
* How exactly is the spectral norm $ \sigma ( \cdot ) $ implemented?
* How exactly does $ \mathcal{L}_{Jdet} $ look like?
* How were the parameters $ \lambda_1, \lambda_2, \lambda_3 $ chosen and what values were used?
* I wonder how spectral normalization compares to other Lipschitz constraints, such as weight clipping used in WGAN.
* I wonder if the presented method is still able to model high frequency components of the deformation field.

**Special Issue:**

no

---

### Meta-Review · Area_Chair_n6qF · 2022-02-14

**Recommendation:** Accept (Oral)
**Confidence:** 5

**Metareview:**

This paper proposed novel approaches to diffeomorphic image registration using residual networks. The reviewers raised some initial questions in their comments, which were addressed by the authors in their rebuttal. All reviewers now agree that the paper is ready for publication at MIDL. Based on their recommendation, I'm happy to accept this work.

---

### Decision · Program_Chairs · 2022-02-28

Accept